# The Role of Network and Identity in the Diffusion of Hashtags

## Abstract

The diffusion of culture online (e.g., hashtags) is theorized to be influenced by many interacting social factors (e.g., network *and* identity). However, most existing computational cascade models model just a single factor (e.g., network *or* identity). This work offers a new framework for teasing apart the mechanisms underlying hashtag cascades. We curate a new dataset of 1,337 hashtags representing cultural innovation online, develop a 10-factor evaluation framework for comparing empirical and synthetic cascades, and show that a combined network+identity model performs better than a network- or identity-only counterfactual. We also explore the heterogeneity in this result: While a combined network+identity model best predicts the popularity of cascades, a network-only model has better performance in predicting cascade growth and an identity-only model in adopter composition. The network+identity model most strongly outperforms the counterfactuals among hashtags used for expressing racial or regional identity and talking about sports or news. In fact, we are able to predict what combination of network and/or identity best models each hashtag and use this to further improve performance. In sum, our results imply the utility of multi-factor models in predicting cascades, in order to account for the varied ways in which network, identity, and other social factors play a role in the diffusion of hashtags on Twitter.

## CCS Concepts

• **Networks** → **Network simulations**; • **Applied computing** → **Law, social and behavioral sciences**; • **Computing methodologies** → *Agent / discrete models*; Network science; Model verification and validation.

## Keywords

hashtags, cascade prediction, cascade evaluation, social network, social identity

**ACM Reference Format:**
Anonymous Author(s). 2025. The Role of Network and Identity in the Diffusion of Hashtags. In *Proceedings of ACM Web Conference 2024 (WWW '25)*. ACM, New York, NY, USA, 14 pages. https://doi.org/XXXXXXX.XXXXXXX

## 1 Introduction

Hashtags are fun. Their flexible meta-linguistic use in social media allows authors to make side-commentary, frame their content within a specific context, vote, or even participate in social movements [48, 62, 71, 77, 91]. Despite their widespread use—e.g., an

estimated 20% of Twitter (now known as $\mathbb{X}$) posts have a hashtag—most hashtags emerge organically, with new hashtags invented regularly by users [60, 82]. While we know much about how hashtags are used, few studies have directly tested the cultural and social forces that shape their adoption at scale. Here, we propose a new framework for teasing apart the mechanisms underlying hashtags cascades as an example of cultural production.

The adoption of cultural products, including hashtags, is theorized to be influenced by multiple interacting social factors [17, 25, 33, 54, 94]. Hashtags tend to spread through *social networks* on Twitter, where users are exposed to a hashtag when a connection tweets it [56, 83]. Additionally, hashtags are often used to explicitly signal some aspect of a user's *social identity*, including their demographic attributes [10, 12, 29, 61]. In these cases, the salient attributes of a user's identity inform the decision to use a hashtag. For example, Sharma [81] theorizes that the spread of hashtags on Black Twitter, a subcommunity discussing Black culture and relevant topics to the Black community, is driven by a combination of network and identity. Adopters are often part of the Black Twitter network and, as such, continued adoption largely occurs within this community because 1) these users are more likely to be exposed to the hashtag, 2) exposed users outside the community tend not to adopt the hashtag if it does not signal their racial identity, which 3) minimizes exposure and adoption outside the community. In other words, in this conceptual model, a crucial part of hashtag cascades in Black Twitter is the interaction between network effects and identity effects. The Twitter network affords users exposure to the hashtag, and each user's racial identity helps determine whether they adopt the hashtag and, therefore, also shapes future exposures.

These interacting network and identity effects are also theorized to play a role in the diffusion of many other types of online cultural products, including sports hashtags [84], hashtags and frames created during the #MeToo online movement [62], branded hashtags [47], neologisms on Twitter [5], and online content related to family planning [66]. However, in spite of numerous conceptual frameworks that describe cultural diffusion as the interaction between network and identity effects, hashtag cascades have primarily been empirically modeled through the lens of social networks alone. For instance, prior work analyzes the effects of different network topologies and contagion processes on cascades [45, 46, 69] and studies how these effects vary by properties like the hashtag's topic and semantics [51, 72]. While some prior literature models how identity is related to hashtag adoption [70, 96], this work often focuses on identity effects in isolation rather than their interaction with network effects. However, as the example of Black Twitter illustrates, the dynamics underlying network-only or identity-only diffusion likely differ from the dynamics when diffusion involves the interaction of network and identity effects. For instance, a user's decision to adopt a hashtag depends on exposure from their network and relevance to their identity, which then shapes future exposure.

In this paper, we present, what is to our knowledge, the first empirical model that explores the joint role of network and identity

effects in the spread of hashtags on Twitter. Using a recently developed agent-based model, we simulate the diffusion of 1,337 hashtags through a 3M-node Twitter network. Agents choose whether to adopt each hashtag based on exposure from the network (Network-only), demographic identity of the hashtag's users (Identity-only), or both (Network+Identity). Our work makes four contributions. We curate a new **dataset** of hashtag cascades, including the initial and final adopters for 1,337 systematically identified hashtags on Twitter that represent the production of novel, popular culture on Twitter (e.g., #learnlife, #gocavs). We also introduce a new **evaluation framework** consisting of ten commonly studied properties of cascades, including their popularity, growth, and adopter composition. Using these assets, we show that a model **integrating network and identity** effects better predicts cascades than models using just one of these factors; the performance improvements are especially pronounced in hashtags signaling regional or racial identity, and those discussing sports or news. Finally, we develop a **predictive model** of when hashtag cascades are best predicted by network alone, identity alone, or network and identity together.

## 2  Related Work

*Production of Novel, Popular Culture.* In online spaces, the ease of content delivery allows a broad set of users to contribute to the creation, rather than simply consumption, of novel culture [60, 78]. In this context, the production of culture entails the creation of symbols (e.g., hashtags) that reflect the values, social structures, and ideologies of its creators [68]. In this sense, hashtags are a cultural product, allowing users to position their tweets in the context of ongoing conversations [48, 65, 82]. The digital studies literature often discusses four key factors related to the dissemination of culture online: how the network of relationships between users allows for the creation of many culturally distinct communities [1, 37, 53, 78, 80], how users define and express their identity online to position themselves within existing communities while also set themselves apart from others [13, 22, 40, 92], how platform design affords the creation and spread of new culture [63, 64], and the types of content created [30, 59]. Our work builds on the literature on cultural production in digital spaces by computationally modeling the effects of two of these factors, network and identity, in the creation and dissemination of hashtags. We introduce a dataset of 1,337 hashtags representing user-created popular culture.

*Modeling Diffusion of Culture Online.* The diffusion of behavior and information online is a topic of significant study. cf. [96], [56], [70], and [83] for recent reviews of this literature. Empirical models often aim to predict some property of the final cascade given some information about its initial adopters. Many such papers adapt models developed to simulate offline behaviors from first principles, including the Susceptible-Infectious-Recovered (SIR) compartment model, the linear threshold model of complex contagion, and stochastic simulations like Hawkes models or Poisson processes. Other papers use deep learning for the predictive task, including graph representation learning and predictive models from features of the network, adopters, and early parts of the cascade. Our work builds on these studies by using a more recent agent-based model of diffusion that accounts for diffusion dynamics particular to Twitter. For instance, by adopting a usage-based instead of adopter-based

model, our framework accounts for frequency effects in the adoption [11, 27]; and by modeling the fading of attention online our model allows for cultural products to stop being used over time (e.g., to model hashtags that are used temporarily and then exit the lexicon) [15]. Using a first-principles model also allows us to test the specific mechanisms associated with network and identity that are encoded in the model – and to explicitly test the *effects* of network and identity in cultural diffusion rather than simply using network or identity features in a predictive model. We also introduce a novel dataset of hashtag cascades and a ten-factor evaluation framework to support future work in this area.

*Social Factors in the Adoption of Hashtags.* Prior work often attributes hashtag adoption to social factors related to *network*, *identity*, *lifecycle*, and *discourse*. Network factors include the position of initial adopters in the social network and simulating the diffusion of innovation through a social network [31, 56]. Identity factors include wanting to join or signal membership to a certain community [65, 70, 93]. Lifecycle factors include the hashtag's growth trajectory [18, 31, 57], and discursive factors include the hashtag's relevance, topicality, and ease of use (e.g., length) [21, 31, 32, 57, 93]. In addition to individual social factors, some conceptual models of diffusion posit that the *interaction* of *multiple* social factors may play a role in the diffusion of hashtags [81, 84]. However, most empirical models of hashtag adoption focus on just one social factor. For instance, [70] notes a number of articles that, separately, describe the effect of "network factors" and "user factors" (e.g., identity) on the propagation of misinformation, but none that describe the effects of both network and user factors. Similarly, [96] lists several papers that model adoption decisions based on either "neighboring relations" (i.e., the network) or "individual/group characteristics" (like identity), but not both. [5] proposes an agent-based model for the adoption of new words online that incorporates both network and identity effects. Our work builds on this prior literature by adapting [5] to empirically model the interaction of two social factors, network and identity, in the diffusion of hashtags.

## 3  Methods

To test the roles of network and identity in the diffusion of hashtags, we use the model specification from Ananthasubramaniam et al. [5] to test whether an agent-based model incorporating network only, identity only, or both network *and* identity best predicts properties of hashtag cascades on Twitter. Although the model is not a novel contribution of this paper, we summarize all key methodological points in this section; the original paper has full details.

### 3.1  Modeling Diffusion of Innovation

Testing our study's hypotheses requires comparing empirical cascades against cascades simulated using the Twitter network and users' demographic identities. In this section, we describe how we produce the needed synthetic cascades.

*3.1.1  Simulation Formalism.* To better simulate the dynamics underlying cultural production, [5] adapts the classic linear threshold model to determine whether each agent will decide whether to use the hashtag depending on prior adoption by other agents. Assume we are given a weighted Twitter network $G$, a vector representing

the identity of each user $i$ in the network $\Upsilon_i$, a hashtag $h$, and a set of initial adopters $A \subset V(G)$ who use $h$ at time $t = 0$. For each edge $(i, j) \in E(G)$, $w_{ij}$ is the strength of the tie and $\delta_{ij}$ is the similarity between the users' identities (i.e., the normalized distance between $\Upsilon_i$ and $\Upsilon_j$). The hashtag can be used to signal an identity represented by vector $\Upsilon_h$, and $\delta_{ih}$ is the similarity between user $i$'s identity and the identity connoted by $h$ (i.e., the normalized distance between $\Upsilon_i$ and $\Upsilon_h$). At each timestep after $t = 0$, agent $i$ has probability $p_{iht}$ of adopting $h$ at time $t$, proportional to:

$$p_{iht} \sim S_h \cdot \delta_{ih} \frac{\sum\limits_{j \,\in\, neighbors \; who \; adopted} w_{ji}\delta_{ji}}{\sum\limits_{k \,\in\, all \; neighbors} w_{ki}\delta_{ki}} \qquad (1)$$

where $S_h$ is a free constant parameter described below.

In the linear threshold model, agents become adopters if the (weighted) fraction of their ego-network that adopt crosses a certain threshold [16]. The model we use relaxes two assumptions to be more relevant for modeling online culture. First, since repeated exposure is important to the adoption of textual innovation [28, 88], this model is *usage*-based, allowing agent $i$ to decide whether to use $h$ at each time step instead of representing adoption as a binary property of the agent (i.e., an agent is either "an adopter" or "not an adopter"). Second, the model uses not only the topology of the social network but also the identity of agents to model the diffusion of innovation. Consistent with prior work on adoption of innovation [9, 16], the network influences each agent's level of exposure to $h$ (the linear threshold-like term in Equation 1). Consistent with prior work on identity performance [24, 35], agents preferentially use hashtags that match their own identity ($\delta_{ih}$) and that are used by demographically similar network neighbors ($\delta_{ij}$).

*3.1.2 Model Parameters.* Each hashtag has a different propensity to be used on Twitter, due to differences in factors like the size of potential audience, communicative need, and novelty [8, 81]. Accordingly, in Equation 1, each hashtag is associated with a different constant of proportionality $S_h$. The $S_h$ parameter is termed *stickiness* because larger values of this parameter bias the model towards higher levels of—or "stickier"—adoption. The stickiness of each hashtag is calibrated to the empirical cascade size (number of uses) using a nested grid search on a parameter space of $[0.1, 1]$ where we first identify the interval of width 0.1 in which the model best approximates the empirical cascade size and then identify the best fitting parameter in that interval using a grid search with step size 0.01. Grid searches are performed using one run of the model at each value of stickiness.

The model has three hyperparameters that apply across all hashtags. These are taken from [5], which tuned the parameters to the empirical cascade size with the same set of users.

*3.1.3 Comparing Network and Identity.* To understand the effects of network and identity, we compare the full Network+Identity model described above against two counterfactuals: 1) the Network-only model, where we simulate the spread of the word through just the network with no identity effects (this is achieved by setting $\delta_{ij} = 1$ and $\delta_{ih} = 1$) and 2) the Identity-only model, where we eliminate the effects of homophily by running simulations on a

configuration model random graph with the same users and degree distribution as the original network.

## 3.2 Network and Identity Estimation

This section elaborates on how network and identity are estimated. Each agent in this model is a user on Twitter who is likely located in U.S.A., based on the geographic coordinates tagged on their tweets [20]. There are 3,959,711 such users in the Twitter Decahose, a 10% sample of tweets from 2012 to 2022. Since we use the same agent identities and network to model the diffusion of all hashtags during this ten-year period, the network and identities are inferred from 2018 data, which is at the midpoint of this timeframe (e.g., identities are from the 2018 American Community Survey and House of Representative elections, the network is inferred from interactions between 2012 and 2018).

*3.2.1 Agent Identities.* In this model, identity includes an agent's affiliations towards 25 identities within five demographic categories: (i) race/ethnicity, (ii) socioeconomic status; (iii) languages spoken; (iv) political affiliation; and (v) geographic location. Each agent's demographic identity is modeled as a vector $\Upsilon \in [0, 1]^{25}$ whose entries represent the proportion of residents in the user's Census tract and Congressional district with each demographic identity. An agent's location is inferred using the geographic coordinates they tweeted from, using the high-precision algorithm from Compton et al. [20]. An agent's political affiliation is the fraction of votes each party got in the agent's Congressional district during the 2018 House of Representatives election. An agent's race, socioeconomic status, and languages spoken are the fraction of the Census tract with the corresponding identity in the 2018 American Community survey. Details on identity calculation are in Appendix C.

*3.2.2 Network.* This study uses a weighted Twitter mutual mention network, which contains ties that are likely to be important in information diffusion. Although Twitter users are exposed to content from more users than they reciprocally mention (e.g., their follower network, public tweets), prior research has shown that the mention network captures edges likely influential in information diffusion [42], while reciprocal ties are often involved in the diffusion of hashtags [73]. The nodes in this network are all agents and there is an edge between agents $i$ and $j$ if both users mentioned the other at least once in the Twitter Decahose sample. The strength of the edge from $i$ to $j$ is proportional to the number of times user $i$ mentioned user $j$ in the sample. Although all ties are reciprocated, the network is directed because the strength of the edge from $i$ to $j$ may not match the strength of the edge from $j$ to $i$. This network contains 2,937,405 users and 29,153,138 edges.

## 3.3 Hashtags

This study models the spread of 1,337 popular hashtags between 2013 and 2022. This section describes how hashtags and their initial adopters and identities are selected.

*3.3.1 Identification.* This paper seeks to study the roles of network and identity in the lifecycle of the *production of novel, user-generated culture*, and we select hashtags that are instances of this phenomenon. First, the hashtag must be a **new** coinage. We aim to model the

spread of hashtags from when they're created, so we include hash-tags that have had low adoption before the data collection window in 2013 (i.e., they were not already in the lexicon) and whose initial adopters we can identify in our data. We also select hashtags that are **well-adopted**. Sufficiently popular hashtags can be considered cultural products, used to allow Twitter users to position their own thoughts in context of a broader conversation [48, 65]. To ensure that the hashtag was popular enough to be considered part of a "broader conversation," we included only hashtags with 1,000 or more uses in our Decahose sample. Finally, we select hashtags that are likely to represent **user-generated** culture. Therefore, hashtags are not common words, phrases, and named entities (e.g., celebrities, movie titles) that appear in WikiData and the dictionary. Instead, they are neologisms or novel phrases that are partly or wholly cre-ated by the community. After this filtering, we were left with 1,337 hashtags. Appendix D.1 details how these criteria were operational-ized. Two authors reviewed 100 hashtags and determined that 84% of them were examples of novel user-generated culture, suggesting a high precision sample (annotation guidelines in Appendix A).

*3.3.2 Hashtag Initial Adopters and Identity.* A hashtag's initial adopters $A \subset V(G)$ are the first ten users who adopted the hashtag (Appendix D.2 has details). Each hashtag signals an identity, deter-mined by the composition of its initial adopters. Initial adopters who are more strongly aligned with a particular identity are more likely to coin hashtags that signal that identity [3]. Accordingly, if the median initial adopter is sufficiently extreme in any given register of identity (in the top 25th percentile of that identity, using the threshold from [5]), the hashtag signals that identity.

## 4 Evaluating Simulated Cascades

When comparing empirical and simulated adoption, researchers often choose to focus on reproducing certain desired properties of a cascade (e.g., common metrics include cascade size, growth, and virality) rather than predicting exactly which individuals will adopt the focal behavior, because there is a high degree of stochasticity in adoption decisions [39, 56, 96]. However, the properties used in the literature vary widely and performing well in one metric is often uncorrelated with performance in another metric. In order to comprehensively study the effects of network and identity on the diffusion of hashtags on Twitter, we develop a framework to analyze a model's ability to reproduce ten different properties of cascades, related to a cascade's popularity, growth, and adopter composi-tion. This requires evaluating models across all ten measures and then combining the ten evaluation scores into a composite *Cascade Match Index (CMI)* to measure the overall performance across the ten measures. To enable error analysis, we do not compare the distribution of properties over all trials; instead we calculate the CMI score for each pair of simulated and empirical cascades and then average errors over all simulations.

For each of the ten metrics, we explain 1) what property of the hashtag is being measured and 2) how comparisons between pairs of simulated and empirical cascades are made.

### 4.1 Popularity

Cascades are often modeled with the goal of understanding the dynamics underlying *popularity* [46, 96]. More popular hashtags

experience high levels of adoption or adoption in parts of the social network that are very distant from the initial adopters, increasing the influence they have on popular culture.

*M1: Level of Usage.* One of the most common metrics used to measure the popularity of a new behavior is simply how often the behavior is used. M1 calculates the number of times a hashtag is used in each cascade, including repeated usage by a user. Comparing simulated and empirical usage requires a measure that operates on a logarithmic rather than a linear scale (e.g., not relative error), because the level of usage could span several orders of magnitude. For instance, if the empirical cascade had 1,000 uses in the Decahose sample (or an expected 10,000 uses on all of Twitter), simulation 1 had 5,000 uses, and simulation 2 had 20,000 uses, a measure like relative error would show that simulation 1 has smaller error than simulation 2 ($|10,000 - 5,000|$ vs. $|10,000 - 20,000|$); however, since cascades often grow exponentially [95], it would be better for both to have the same magnitude of error since one is half as big and the other is twice as big as the empirical cascade. Therefore, we compare the ratio of simulated to empirical usage on a logarithmic scale $|log(\frac{M1_{sim}}{10 \cdot M1_{emp}})|$, henceforth referred to as the *log-ratio error*. We compare $M1_{sim}$ to $10 \cdot M1_{emp}$ because the empirical cascades are drawn from a 10% sample of Twitter and, therefore, we expect $M1_{emp}$ to be 10 times larger on all of Twitter.

*M2: Number of Adopters.* In addition to the level of usage, another popular way of measuring popularity is the number of unique adopters in a cascade. M2 looks at the number of unique users *in the downsampled cascade* who adopted each hashtag. Unlike M1, M2 does not consider repeated usage and may be much lower than M1 when a cascade experiences a high volume of usage by a small group of users (e.g., for niche cascades that are really popular among a small group of users); however, in many cases, M1 and M2 are likely to be correlated. Since, like M1, the number of adopters also scales exponentially, comparisons between empirical and simulated cascades are made using the log-ratio error.

*M3: Structural Virality.* Another metric for a hashtag's popularity is how deeply it has permeated the network, or its structural virality [34]. With unknown initial adopters, structural virality is measured as the mean distance between all pairs of adopters (the Wiener index). However, as initial adopters are known in our case, structural virality is defined as the average distance between each adopter and the nearest seed node. Since distances are usually between 3 and 12 hops [52], comparisons between simulated and empirical structural virality are made using relative error $\frac{|M3_{sim} - M3_{emp}|}{M3_{emp}}$.

### 4.2 Growth

In order to understand how hashtags become viral, many studies look not just at the popularity of a hashtag but also how its adoption shifts over time [17, 76]. There are a number of commonly studied properties of cascades that measure how they grow.

*M4: Shape of Adoption Curve.* The shape of a hashtag's adoption curve (or the number of uses over time) is indicative of different mechanisms that may promote or inhibit a cascade's growth [67, 76]. M4 is modeled by splitting both the simulated and empirical time series into $T$ evenly-spaced intervals, where $T$ is the smaller of

a) the number of timesteps in the simulation and b) the number of hours in the empirical cascade. To make the empirical curve comparable to the simulated curve, we first truncate the adoption curve's right tail once adoption levels remain low for a sustained period of time, to match the simulation's stopping criteria. We compare the empirical and simulated curves using the dynamic time warping (DTW) distance between them.

*M5: Usage per Adopter.* The growth patterns of hashtags vary based on how often each user posts a hashtag [27]. M5 calculates the average number of times each adopter used the hashtag. Simulated and empirical cascades are compared with relative error.

*M6: Edge Density.* The structure of the adopter subgraph of the network often reflects how a cascade grows and spreads through the network [4]. In particular, to model connectivity, M6 is operationalized as the normalized number of edges, or edge density, within the adopter subgraph.[1] Since edges in the adopter subgraph can be very sparse or very dense and these scenarios change the number of edges by several orders of magnitude, the empirical and simulated edge densities are compared using the log-ratio error.

*M7: Growth Predictivity.* In many cases, it is useful to be able to predict how big a cascade will become based on a small set of initial adopters [18, 46, 55]. In order to test how well each model achieves this task, we attempt to predict the size of each empirical cascade based on the characteristics of the first 100 adopters in each simulation using a multi-layer perceptron regression with 100 hidden layers, an Adam optimizer, and ReLU activation. Predictors include a set of 711 attributes from Cheng et al. [18] that are not directly used by our models: the timestep at which each of the first 100 adopters used the hashtag; the degree of each adopter in the full network and adopter subgraph (note that the identity-only model preserves degrees of each agent); and the age and gender of each adopter, inferred using Wang et al. [89]'s demographic inference algorithm, etc. Simulated and empirical cascades are compared using the relative error of the predicted cascade size.

### 4.3 Adopters

In addition to modeling popularity and growth, a body of research has studied how certain subpopulations come to adopt new culture [6, 19]. We identify a set of three measures of how well a simulated cascade reproduces the composition of adopters.

*M8: Demographic Similarity.* New culture is often adopted in demographically (e.g., racially, socioeconomically, linguistically) homogenous groups. This may occur when the cultural product is explicitly signaling an affiliation with the demographic identity (e.g., #strugglesofbeingblack) or by convention [2, 23, 86, 87]. We compare the distribution of agents demographic attributes in adopters from empirical and simulated cascades. Since there are many demographic attributes, we construct a one-dimensional measure of these attributes using a propensity score. This propensity score is the predicted probability obtained by regressing the demographic

---

[1]Another commonly studied property of the adopter subgraph is the number of connected components. We chose not to use the number of connected components because the corresponding error was reasonably correlated with edge density, so they didn't seem like sufficiently different measures; additionally, unlike edge density, the connected components often change dramatically after downsampling.

attributes on a binary variable indicating whether the user is from the simulated or empirical cascade. This propensity score has two important properties: 1) users that are adopters in both cascades will not factor into the construction of the propensity score since they are represented as both 1's and 0's in the logistic regression; and 2) if the empirical and simulated adopters have similar demographic distributions, the propensity scores of adopters in the empirical cascade will have a similar distribution as the propensity scores of adopters in the simulated cascade [74]. The differences in demographics between simulated and empirical cascades is measured using the Kullback–Leibler (KL) divergence of the distribution of the empirical adopters' and simulated adopters' propensity scores.

*M9: Geographic Similarity.* Another property of interest is whether a model can reproduce where adopters of a hashtag are located in U.S.A. [26, 41]. The location of adopters is modeled as a smoothed county-level distribution of the fraction of users in the county who adopted the hashtag. Geographic similarity is measured as the Lee's $L$ spatial correlation between the spatial distributions of empirical and simulated usage [5, 50].

*M10: Network Property Similarity.* Another property of cascades is the position of adopters within the network [43, 90]. We calculate each user's position in the network along four relatively low-correlated (Pearson's $R < 0.5$) network properties, including PageRank, eigencentrality, transitivity, and community membership (using the Louvain community detection algorithm [14]). Similar to M8, we represent the adopters' network positions using a propensity score, and compare the empirical and simulated adopters' propensity scores using KL divergence.

### 4.4 Composite Metric

For a more holistic evaluation, we construct a composite Cascade Match Index (cmi) encompassing all ten metrics. The cmi is calculated by z-scoring each metric M1-M10 then averaging z-scores over all ten metrics. See Appendix B for details. The ten measures comprising the cmi are overall poorly correlated with each other (Figure S1), suggesting that M1-M10 do, in fact, measure distinct properties of the cascade and are not redundant.

## 5 Network and Identity Model Different Attributes of a Cascade

To test our hypothesis, we simulate hashtag cascades using the Network+Identity, Network-only, and Identity-only models, and determine which one best matches properties of empirical cascades.

### 5.1 Experimental Setup

For each of the 1,337 hashtags and three models, we 1) seed the model at the hashtag's initial adopters, 2) fit the stickiness parameter, 3) run five simulations at this parameter, and 4) compare properties of the simulated and empirical cascades. Then we construct the cmi and compare values across the three models.

### 5.2 Results

Figure 1 shows that the Network+Identity model outperforms the Network-only and Identity-only counterfactuals on the composite

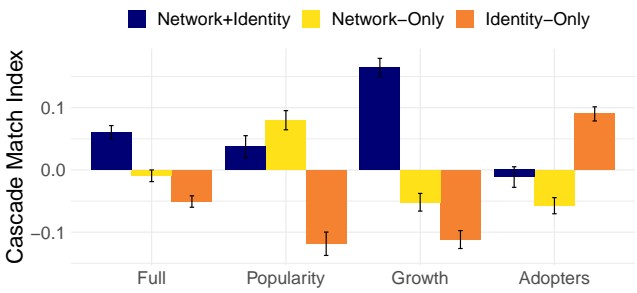

**Figure 1: The Network+Identity model outperforms the Network-only and Identity-only baselines. Models evaluated on the full cmi and just the subset of indices corresponding to popularity, growth, and adopter characteristics. Higher cmi scores corresponds to better performance.**

cmi—suggesting that, on the whole, hashtag cascades are best modeled using a combination of network and identity. However, our results also suggest that, while models involving both network and identity are most performant overall, there is important variation in what social factors are required for different properties of hashtag cascades. Thus, while incorporating network and identity leads to the highest overall performance, the network-only or identity-only model may be a better choice if certain features are the target.

As shown in Table S1, the Network+Identity had the top performance on a larger number of individual metrics (5 of 10) than the Network-only (2) or Identity-only (3) models. Overall, the Network-only model tended to perform best on popularity-related metrics; it had the highest score on M2 and M3, as well as a higher score on a composite index of the three growth-related measures. On the other hand, the Identity-only model tends to perform better on adopter-related metrics, while growth-related metrics were best modeled by a combination of both network and identity. Network+Identity performed best on growth-related metrics and second best in the other types of metrics. A possible explanation for the heterogeneity in performance is that different mechanisms are responsible for different properties of cascades.

## 6 Network and Identity in Context

The diffusion of hashtags specifically, as well as the process of cultural production more generally, varies across contexts. For instance, hashtags with demographically homogenous initial adopters are more likely to be used to signal identity [3, 84]. Additionally, hashtags have different patterns of diffusion depending on their topic or semantic context [51, 72]. The goal of this section is to understand whether information about the hashtag and its initial adopters are associated with model performance.

### 6.1 Experimental Setup

In order to understand the relationship between the context in which each hashtag was coined and the role of network and identity, we run a linear regression to test the association between the cmi and several properties of the hashtag. As shown in Equation 2, we estimate the effect of each covariate $c_i$ on the cmi of each model.

$\beta_i$ are the regression coefficients, where $\beta_1$ estimates the effect of the first covariate on cmi in the Network+Identity model, $\beta_1 + \beta_1^N$ estimates the effect in the Network-only model, etc. Our regression estimates the effect of each property after controlling for all other properties (e.g., the effect of racial similarity in initial adopters is independent of the effect of their geographic proximity, even though these two factors are correlated).

$$CMI \sim \beta_0 + \sum_i \beta_i c_i + \sum_i \beta_i^I c_i * 1_{Id-only} + \sum_i \beta_i^N c_i * 1_{Net-only} \quad (2)$$

Covariates are four sets of properties of the hashtag's context (the distribution of each property is in Figure S2):

*Topic.* The topic of a hashtag (e.g., whether it is related to sports, pop culture, or some other subject matter) may be associated with the extent to which different mechanisms like network and identity play a role in its diffusion [72]. Therefore, we include each hashtag's topic, measured using the model from Antypas et al. [7], as a covariate in Equation 2. Appendix E lists all the topics used.

*Communicative Need.* Properties of hashtag cascades may also be attributable to differences in communicative need for a hashtag [44, 51, 85]. Ryskina et al. [75] quantified communicative need using two measures: 1) semantic sparsity, or how many similar hashtags exist in the lexicon when the focal hashtag was introduced (a hashtag in a sparse space may be in higher demand since there are fewer hashtags that can serve the same function); and 2) semantic growth, or the growth in the semantic space over time (a hashtag in a high-growth space may be in higher demand since it serves a purpose of increasing popularity). For instance, a hashtag like #broncosnation (signifying support for the city of Denver's local football team) has low semantic sparsity, because many cities had similar sports hashtags when it was coined; it also has low semantic growth because, while sports team hashtags are popular, the use of these sorts of hashtags has remained fairly stable over time. Appendix E has details on how these measures are operationalized.

*Identity.* As described in Section 3.3, each hashtag's identity is based on the demographics of the first ten adopters. Since the identities of early adopters may influence the perception of the hashtag [3], and since having more homogenous initial adopters may lead to stronger perceptions, covariates include the mean pairwise similarity of the ten initial adopters within each component of identity (location, race, SES, languages spoken, political affiliation).

*Initial Network Position.* Another factor in a hashtag's diffusion is where in the network the hashtag is introduced [43, 90]. For instance, more central initial adopters may be able to spread the hashtag more broadly because of their influence. Therefore, we include the median initial adopter's eigencentrality as a covariate.

### 6.2 Results

Figures 2-3 show the regression results; in Figure 2, the y-axes (and in Figure 3, the x-axis) plot the predicted cmi for each model corresponding to different levels of each covariate, holding all other covariates constant. In all conditions, the Network+Identity model performs as well as or better than the other models. This suggests

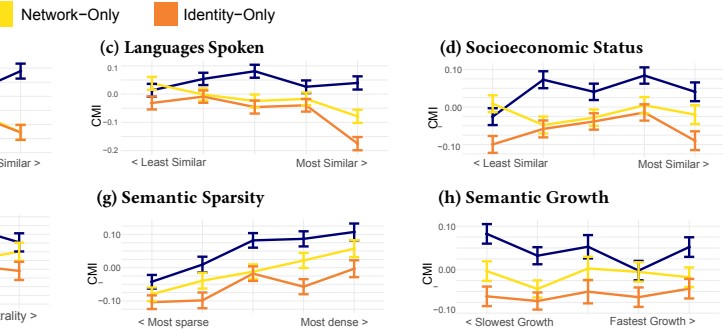

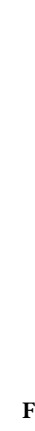

Figure 2: The comparative advantage of modeling cascades using both network and identity is highest when a) initial adopters are located very close to each other; b) have a high degree of racial similarity; c-f) have a moderate degree of linguistic, socioeconomic, and political similarity and eigencentrality; g) hashtags convey a similar meaning as a moderate number of other hashtags, and h) their meaning is not becoming increasingly popular over time. Effects are estimated by running a regression, controlling for other variables related to the hashtag's context.

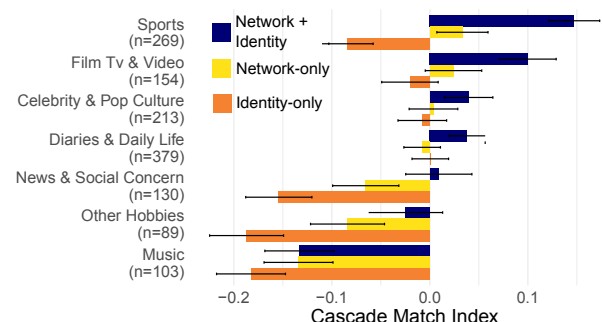

Figure 3: Although the Network+Identity model never underperforms the others, the relative advantage of the Network+Identity model varies by the topic of the hashtag. Effects are estimated by running a regression, controlling for other variables related to the hashtag's context.

that the conclusions from Section 5—that network and identity better predict cascades together than separately—are robust.

The Network+Identity model tends to outperform the other models in cases where there is a theoretical expectation that network and identity would each contribute to the underlying diffusion mechanism. For instance, when initial adopters have a high level of racial similarity, the Network+Identity model's performance improves while other models get worse (Figure 2b); this is consistent with the theoretical framework of Sharma [81], where hashtags used to signal racial identity on Black Twitter diffuse via a mechanism that combines network and identity. A similar mechanism is propose for sports hashtags, where only fans of a specific team adopt the hashtag which shapes exposure in the Twitter network [84]. Similarly, regional hashtags may require network and identity to constrain adopters to the local area [79]; consistent with this expectation, the Network+Identity model has its strongest comparative advantage among hashtags that promote regional culture, including sports hashtags (which often express support for local teams) and hashtags whose initial adopters are located near each

other (Figure 2a,3). The Network+Identity model also has the best performance on geographic distribution of adoption, suggesting a connection between this model and the ability to predict geographic localization [5, 49]. Similarly, hashtags related to certain topics—sports, film/TV/video, diaries/daily life, and news/social concern—tend to be better modeled by the Network+Identity simulations than others (Figure 3). These hashtags are often used in conversations that involve identity signaling in order to take a stance (e.g., sharing their opinion on issues of social concern, their favorite TV show, and aspects of daily life) [29].

Additionally, the Network+Identity model may outperform the Network-only and Identity-only models because hashtags that diffuse via two mechanisms are more likely to become *popular* than hashtags diffusing via just one [36, 38]. For instance, the Network+Identity model outperforms baselines among very slow- or very fast-growing hashtags, but not among hashtags with moderate growth (Figure 2f-h). Similarly, the model has its highest comparative advantage when initial adopters are moderately central. In cases of extreme growth or moderate initial adopter centrality, hashtags that diffuse via multiple mechanisms (network and identity) may be overrepresented in our sample of popular hashtags.

Finally, the Network+Identity model often has its strongest comparative advantage when the Network-only or Identity-only models perform well. To test this, we regress the Network+Identity model's CMI score in each trial on the counterfactual Network-only and Identity-only CMI scores, and find a strong, positive association for both counterfactuals (regression coefficients of 0.31 and 0.21 respectively, $p < 10^{-16}$). Therefore, even when single-variable models have relatively high performance, combining multiple social factors can lead to improvements.

## 7 Selecting Among Models

Figure 1 suggests that the mechanisms underlying the diffusion of hashtags are likely heterogeneous: most hashtags are best modeled by a combination of network and identity, but some are better modeled by network alone or identity alone. Moreover, on the whole, the Network+Identity model had the highest score on the CMI in 42% (2,791) of trials, while the Network-only model had the highest

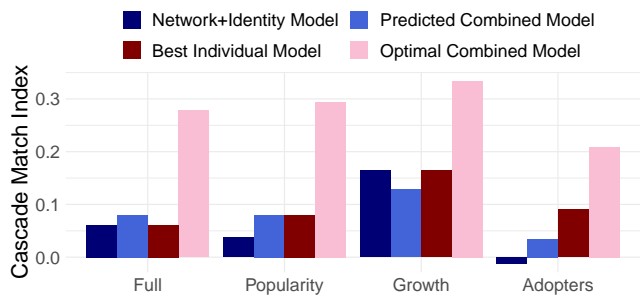

**Figure 4: A *combined model* that selects among the three models does better than the Network+Identity model alone.**

score in 30% (1,992) and the Identity-only model in 28% (1,902) of trials. When we select the model that has the highest score on the CMI for each trial (we'll call this the *optimal combined model*), the average score on the CMI improves from 0.06 in the Network+Identity model to 0.27 with the optimal combined model (Figure 4, pink vs. dark blue bars); this is comparable to the Network+Identity model's improvement over the Identity-only model (0.21 vs. 0.16 points). Identifying whether network and/or identity best predicts a given hashtag's cascade can lead to significant predictive gains. Since our goal is to produce a unified model that reproduces *all* properties of cascades, one option is to create a *predicted combined model* that uses features of the hashtag and early adopters to decide whether to use network or identity or both, instead of an *optimal combined model* where the model selection is performed post-hoc.

Since there are associations between the characteristics of the hashtags and the relative performance of the three models, we develop a *predicted combined model* that uses these characteristics to determine whether network alone, identity alone, or both together would perform best on the CMI. Using the features described in Section 6.1, we trained a random forest classifier to predict whether each hashtag would be best predicted by the Network+Identity, Network-only, or Identity-only model. Predictions were obtained using a repeated 5-fold cross-validation.

The random forest classifier weakly outperforms a baseline that always selects the Network+Identity model (0.44 vs. 0.41 accuracy); the predicted combined model significantly outperforms the Network+Identity model on the CMI (Figure 4, light blue bars), suggesting that the classifier may be picking out examples of hashtags that are "obviously" or "easily" identifiable as being better-modeled by network or identity alone *and* where the single-variable models are associated with significant predictive improvements over the Network+Identity model. This *predicted combined model* achieves its gain in performance by better reproducing properties related to popularity (where it equals the Network-only model's performance) and adopter characteristics, and trading off slightly lower performance on the growth-related measures (Figure 4, comparing light and dark blue bars). These results suggest that the initial characteristics of cascades can, in some cases, signal the driving mechanism behind the hashtag's diffusion and therefore the best model to estimate the cascade.

## 8 Discussion

Our work suggests that modeling cultural production requires explicitly incorporating the role of multiple social factors in the process of diffusion. This study examines the role of network and identity in the diffusion of novel hashtags on Twitter. In order to test the roles of network and identity in diffusion, we evaluate whether a model containing network and identity better reproduces properties of each hashtag's cascade than models containing just network or just identity—and whether this holds across different types of hashtags. The results support our hypothesis from three standpoints. First, the model with both identity and network better reproduces an aggregate of cascade properties than models with identity or network alone. Second, many individual properties are also better modeled with network and identity together. Third, these findings are true across many different types of hashtags (different topics, identities, etc.). These findings are significant because most existing work has focused on the effects of single factors (e.g., network or identity) rather than creating a model that combines multiple social factors to explain the spread of culture. Our work suggests that there is value in adding this extra complexity.

Our analysis also reveals that there is important heterogeneity in the roles network and identity play in cultural production. For instance, network structure does a worse job modeling the adopter composition of cascades, while identity underperforms at modeling a cascade's popularity. Additionally, there are several contexts where the network and identity likely offer non-duplicative conditions for diffusion or jointly confer some selective advantage to new hashtags. For instance, hashtags related to racial or regional culture, sports, and news. Under these conditions, it is especially important for models of cascades to combine both factors.

Our analysis has limitations that can be addressed by future work: Our model only considered network and identity, and not other relevant social factors (e.g., the type of relationships between users). This limitation could be responsible for some heterogeneity in performance. However, such factors are difficult to model at scale and, thus, were outside the scope of the paper. Additionally, in the interest of parsimony, our model did not incorporate many factors unrelated to network and identity that are known to influence diffusion (e.g., structural diversity, correlated diffusion).

The Network+Identity model always used both network and identity. We presented a first step towards developing a *combined model* that selects which features would work best for each hashtag. However, future work could likely improve upon this initial model. In order to facilitate future work, we release a database of the 1,337 hashtags included in this study, which were coined between 2013 and 2022, used frequently, and likely to represent user-created culture; using a 10% sample of Twitter, we develop a database of each hashtag's adoption and a rich set of features like the hashtag's topic, embedding, communicative need, and the identities of adopters. Based on a comprehensive literature review, we identify ten frequently-modeled properties of cascades related to their popularity (e.g., cascade size), growth (e.g., shape of the growth curve), and adopter composition (e.g., demographic similarity) and release a composite CMI that compares empirical and simulated cascades across all ten properties.

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

# Appendix

## A Annotation Prompt

In order to determine whether procedure in Section 3.3 returns hashtag that are relevant to our study, two authors used the following prompt to test a sample of 100 hashtags returned by the procedure:

**Would the coining of this hashtag be an example of the production of novel culture (Yes/No)?** *In this case, cultural production is the process of creating and disseminating new, innovative culture. While "culture" is a broad term, our definition excludes hashtags that make reference to entities by their official name (e.g., a person by their full name or stage name, a location, a song title), common phrases, and single dictionary words, since those hashtags do not seem innovative. However, the following types of hashtags can and should be considered examples of cultural production, because their existence requires innovative choices and combinations of words: nicknames or fan-created names for entities, unusual combinations*

|  | Network+Identity | Network-only | Identity-only |
|---|---|---|---|
| M1. Level of Usage | **3.10 / 0.09** | 3.21 / 0.01 | 3.37 / -0.10 |
| M2. # Adopters | 1.30 / -0.02 | **1.09 / 0.15** | 1.42 / -0.13 |
| M3. Structural Virality | 0.15 / 0.05 | **0.14 / 0.07** | 0.19 / -0.12 |
| M4. Shape of Adoption Curve | 0.12 / 0.05 | 0.13 / -0.16 | **0.11 / 0.12** |
| M5. # Uses per Adopter | **2.18 / 0.14** | 2.49 / -0.11 | 2.39 / -0.02 |
| M6. Adopter Connectedness | **1.83 / 0.29** | 2.08 / 0.11 | 2.80 / -0.40 |
| M7. Growth Predictivity | **1.83 / 0.12** | 2.08 / 0.01 | 2.80 / -0.04 |
| M8. Demographic Difference | 0.98 / -0.31 | 0.62 / 0.10 | **0.52 / 0.22** |
| M9. Geographic Difference | **0.09 / 0.32** | 0.03 / -0.12 | 0.03 / -0.16 |
| M10. Network Difference | 0.31 / 0.04 | **0.33 / 0.16** | 0.26 / -0.20 |

**Table S1: The performance of each model on each metric in our CMI. This includes the raw comparison (e.g., log-ratio error, relative error, similarity score) and normalized comparison (z-score) for each measure.**

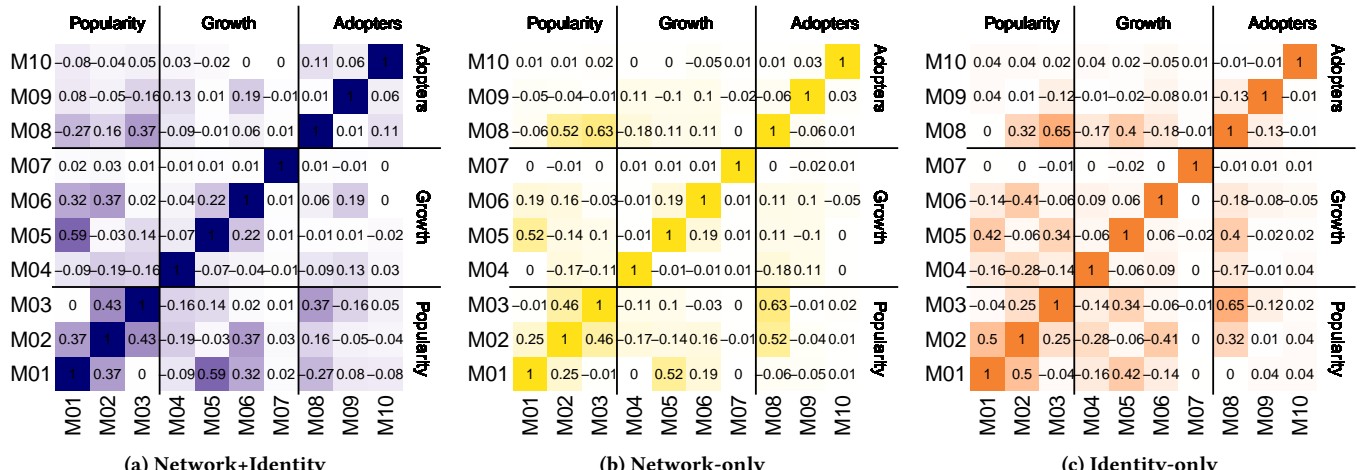

(a) Network+Identity  (b) Network-only  (c) Identity-only

**Figure S1: Correlations between cascade evaluation measure M1–M10 are relatively low, suggesting that these capture distinct properties that can be effectively combined by the CMI.**

*of dictionary words, slogans, and acronyms. Examples of hashtags to say 'Yes' to: #goravens, #rio2016, #votefreddie, #blacklivesmatter, #myboyfriendnotallowedto, #incomingfreshmenadvice*

## B  Constructing the Cascade Match Index

Since M1-M7 are compared using a measure of distance or error (i.e., closer to 0 is better) and M8-M10 are compared using similarity scores, we convert M1 - M7 from difference scores into similarity scores by taking their additive inverse. This means that higher values of the CMI correspond to better fit between empirical and simulated cascades. Additionally, since each measure is on a different scale, we standardize all similarities using a z-score; to facilitate cross-model comparisons, z-scores are calculated across all three models (Network+Identity, Network-only, Identity-only) rather than within each model to allow for cross-model comparison. Finally, since model parameters are calibrated to the cascade size, and since empirical cascades (which came from the Twitter Decahose) are expected to be 10% the size of simulated cascades, we downsample the larger cascade to match the size of the smaller one for properties M2 - M10 (e.g., if the simulated cascade ends

up being 10 times bigger than the empirical cascade, we randomly sample 10% of the simulated cascade and compare that downsampled cascade to the empirical cascade). This downsampling ensures that the comparison between the empirical and simulated cascade is independent of size—e.g., that certain models do not better match properties because they were easier to calibrate to the correct cascade size.

## C  Agent Identity

Each agent's demographic identity is modeled as a vector $\Upsilon \in [0, 1]^{25}$ whose entries represent the proportion of residents in the user's Census tract and Congressional district with these different demographic identities.

A user's geographical lat/lon coordinates are the geographic median of the geolocations they disclosed in their tweets. To ensure high precision, we select only Twitter users with five or more GPS-tagged tweets within a 15km radius, so that we have high certainty about their location. This procedure uses conservative thresholds for frequency and dispersion, and has been shown to produce highly precise estimates of geolocation [20]. This precision may come at

the cost of excluding some users from the study, but we chose a high-precision approach because we agree that it is extremely important to get the correct location so the rest of the identity is correct.

We model each agent's identity as consisting of several components (components related to race: Black, Latino, white, etc.; components related to SES: below poverty line, receive SNAP benefits, less than high school education, unemployed, etc.). Each component of agent identity varies continuously between 0 and 1, where agents closer to 0 affiliate weakly with that identity and agents closer to 1 strongly identify with the register. We infer each component of identity based on the demographic composition of the agent's Census tract and Congressional district. For instance, we infer the agent's race, SES (poverty line, SNAP usage, education, and laborforce status), and language spoken at home, based on the fraction of the agent's Census Tract identifying with each option in the 2018 American Community Survey. The agent's political identity is inferred based on the fraction of the agent's Congressional District that voted for a particular political party in the 2018 House of Representatives election.

Note that each component of an agent's identity is static over time. We make this assumption because many of the demographic attributes have remained fairly correlated over time; e.g., the Spearman rank correlation of each attribute across the 72K census tracts is between 0.85 and 0.96 across the 10 years in our study. Therefore, we would not expect invariant identity to have large impacts on our conclusions. Additionally, within the set of identity features used in our model, such as race, education, or languages spoken at home, most individuals would likely not significantly change any of these aspects with the course of our simulations ( 10 years in total).

## D Hashtags

### D.1 Hashtag Identification

We systematically select hashtags from the Twitter Decahose sample between January 2012 and December 2022. First, we collect all tweets from the Decahose sample that were posted by the 2,937,405 users in our network. These tweets contain 198,988 hashtags that were used at least 100 times. Next, we filter these hashtags, as follows:

(1) **Popularity:** To limit our study to hashtags that eventually became popular, we eliminate 116,477 hashtags that were used fewer than 1,000 times between 2013 and 2022. Frequencies are counted without considering case. While some studies may also consider less popular hashtags, we eliminate these because many of the properties we're interested in can't be calculated or are too noisy on small cascades.

(2) **Novelty:** To limit our study to newly coined hashtags, we eliminate 77,134 hashtags that were used more than 100 times in 2012 (e.g., #obama2012, #sup, #sobad, #sandlot).

(3) **Innovativeness:** To ensure the hashtag represents production of novel culture (e.g., it is not a reference to some named entity, a common phrase, or a dictionary word), we eliminate 3,144 hashtags that were entries in the Merriam Webster English-language dictionary (e.g., #explore, #dirt) or in Wikidata, a repository of popular named entities

and phrases (e.g., #domesticviolence, #billcosby, #interiordesign). Since hashtags cannot contain certain characters that might appear in the dictionary and Wikidata (e.g., spaces, apostrophes, periods), we replaced these characters with both spaces and underscores to ensure that we eliminate hashtags using these different conventions. Two authors reviewed a sample of 100 of these hashtags and determined that 84% of them were examples of user-generated cultural production, rather than references to entities, dictionary words, or other non-cultural or existing cultural references (annotation guidelines in Appendix A).

(4) **Presence of Seed Nodes:** To ensure that the hashtag was coined between 2013 and 2022, we eliminate 896 hashtags whose cascade began before 2013 (e.g., #theedmsoundofla, #southernstreets, #rastafarijams). The procedure to identify seed nodes is described in Section D.2.

After this filtering, we were left with 1,337 hashtags.

### D.2 Initial Adopters

Each cascade's initial adopters are the users whose adoption of the hashtag 1) was likely not influenced by prior usage on Twitter and 2) likely influenced future adoption of the hashtag. To identify these users, we first find instances where each hashtag had a period of contiguous usage, by looking for periods of time when the hashtag was used at least 100 times in the Decahose sample (i.e., likely at least 1,000 times overall) with less than a month's gap between uses. We assume that the cascade starts during the first period where the hashtag was used more than 1,000 times, since prior work has shown that any usage before this start date is likely unrelated to the cascade as it was used too infrequently for users in the cascade to have a high likelihood of adoption [5]; and adopters after the start date are likely to remember the usage in this first period because of its high frequency [51, 58]. The hashtag's initial adopters are the first ten users to use the hashtag after the start date.

## E Hashtag Characteristics

### E.1 Topic

We define a hashtag's topic as the most frequent topic of the tweets it appears in, where tweet topics are inferred using Antypas et al. [7]'s supervised multi-label topic classifier. From the original set of 23 topics, we combine categories containing fewer than 50 hashtags into other categories that they most frequently co-occur with (e.g., Learning & Educational with Youth & Student Life), and end up with seven categories: diaries and daily life (379 hashtags, e.g., #relationshipwontworkif, #learnlife, #birthdaybehavior), sports (269 hashtags, e.g., #seahawksnation, #throwupthex, #dunkcity), celebrity and pop culture (213 hashtags, e.g., #freesosa, #beyoncebowl, #kikifollowspree), film/TV/video (154 hashtags, e.g., #iveseeneveryepisodeof, #betterbatmanthanbenaffleck, #doctorwho50th), news and social concern (130 hashtags, e.g., #impeachmentday, #getcovered, #saysomethingliberalin4words), music (103 hashtags, e.g., #lyricsthatmeanalottome, #nameanamazingband, #flawlessremix), and other hobbies (89 hashtags, e.g., #camsbookclub, #amazoncart, #polyvorestyle).

## E.2 Semantic Sparsity and Growth

Semantic sparsity and growth are measured as follows: Each hashtag's 250-dimensional embedding is constructed by training the word2vec algorithm over a window of 5 tokens and 800 epochs; in order to ensure that the hashtags in our study have high enough token frequency to be included in the final model, word2vec was trained on all tweets containing the 1,337 hashtags in our sample and a random sample of 20 million other tweets containing hashtags in our Twitter Decahose sample. Using the resulting word embeddings, semantic sparsity is the number of hashtags that were used in similar contexts at the time when the hashtag was coined (similarity means the cosine similarity of the embeddings is at least 0.3,[2] representing the *supply* of similar hashtags) and the semantic growth is the Spearman rank correlation between the frequency of all tokens that are similar to the hashtag and the month (where a correlation of 1 means that words that are similar to the hashtag are becoming more popular over time, and 0 means the hashtag is used in contexts of static popularity).

Received 14 October 2024; revised 14 October 2024; accepted 14 October 2024

---

[2]The threshold of 0.3 is slightly lower than the threshold of 0.35 used in the original paper, so that more hashtags have neighbors.

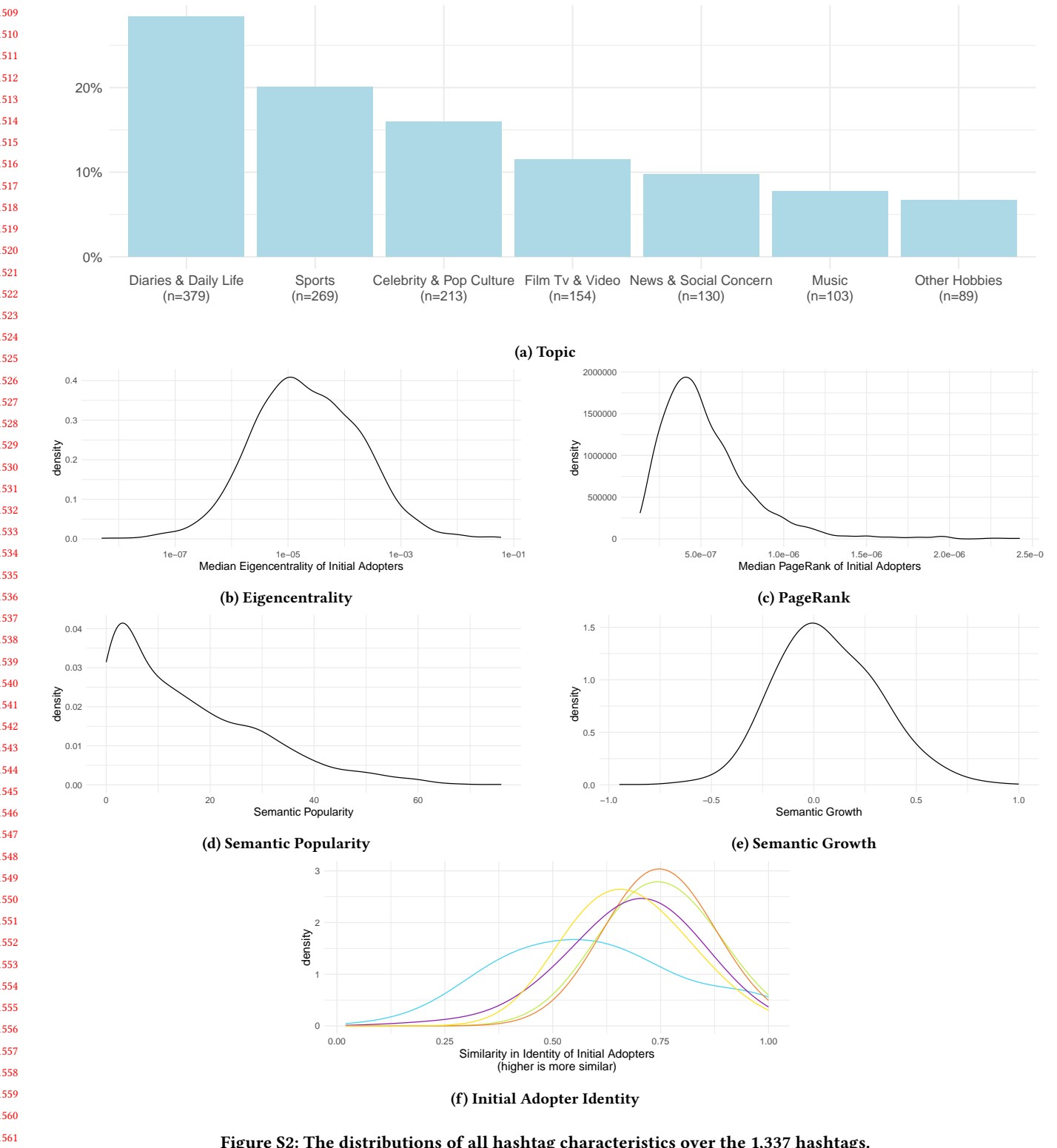

Figure S2: The distributions of all hashtag characteristics over the 1,337 hashtags.

