# OpenReview forum: "Roles of Network and Identity in Hashtag Diffusion"
_ACM.org/TheWebConf/2025/Conference — WWW 2025 Poster_

### Official Review · Reviewer_14Yp · 2024-11-30

**Novelty:** 5
**Technical Quality:** 4

**Review:**

This paper presents a novel approach to modeling and predicting topic diffusion in social networks by integrating network and identity effects, offering a fresh perspective on propagation inference. The work stands out for its originality and its potential significance in enhancing our understanding of multi-factor interactions in online cultural diffusion. However, certain aspects of the paper could benefit from additional clarity and depth of analysis.

Strengths:
	1.	Integrating identity and network effects for propagation inference is an innovative contribution.
	2.	The authors rigorously validated the effectiveness of integrating personal and network information from multiple perspectives within the network model, enhancing the credibility of their approach.

Weaknesses:
	1.	The meaning and interpretation of the CMI metric are unclear, especially regarding the significance of its positive and negative values and their absolute magnitudes. Reintroducing this metric in the context of experimental results is necessary for better comprehension.
	2.	While the combination of identity and network effects is interesting, the authors do not sufficiently discuss the research challenges associated with this integration, leaving the work feeling more like a validation of existing theories rather than an advancement addressing these challenges.
	3.	The relationship between the simulated model and real-world propagation is unclear. Questions such as how learnable parameters in the model are trained and how the validity of the simulated propagation model is guaranteed remain unanswered.

Overall Assessment:
The work is highly original and contributes meaningfully to social network analysis. However, its impact could be further enhanced by addressing gaps in clarity and providing deeper insights into the challenges and validity of the proposed approach.

**Questions:**

1. What does the CMI metric represent? When the CMI takes positive values, what does the magnitude of its absolute value indicate? Similarly, when the CMI takes negative values, what does the magnitude of its absolute value signify? It is recommended that the meaning of CMI values be reintroduced in the context of experimental results when they are first discussed.
2. While combining identity and network effects is intriguing, what research challenges arise during this integration? The authors do not provide a detailed discussion on this point. Overall, the work appears to validate existing propagation theories rather than address the challenges inherent to this task with effective solutions.
3. What is the relationship between the simulated model and real-world propagation? I find this point a bit confusing and unclear. The simulated model should involve some learnable parameters, but how are these parameters learned? Additionally, how is the validity of the simulated propagation model justified?

**Reviewer Confidence:**

3: The reviewer is confident but not certain that the evaluation is correct

**Scope:**

2: The connection to the Web is incidental, e.g., use of Web data or API

---

### Official Review · Reviewer_CwtY · 2024-12-02

**Novelty:** 5
**Technical Quality:** 5

**Review:**

Summary:

This paper introduces a ten-factor evaluation framework to analyze and predict hashtag diffusion. By integrating both network and identity factors, the study simulates the diffusion process and evaluates how these mechanisms interact to shape the adoption of hashtags. The study compares three models: Network-only, Identity-only, and Network+Identity, and demonstrates that combining network and identity factors improves predictive accuracy. Additionally, the paper explores whether a combined model could be optimized to predict the best-performing mechanism.

Highlights:

The paper builds on well-established methodologies from prior literature  and each chosen feature is carefully justified.

The introduction of the CMI is novel and useful.

Overall, the methodology is solid.

Areas of improvement:

The methodology adopted from [5] (a referenced prior paper on new word adoption) is reasonable, but the current paper does not sufficiently explain key components (e.g., the 3 hyperparameters mentioned on Line 279, and I assume they are Q,r,θ from [5]). A reader unfamiliar with [5] would struggle to understand the methodology and assumptions fully. Information should be added to Appendix.
The process of inferring demographic information (e.g., race, location, political affiliation) is unclear, especially given the high likelihood of missing geolocation data in Twitter's Decahose.

There is insufficient detail on how the data was constructed. For instance, it is unclear how demographic information was inferred, especially given the likely high volume of missing geolocation data. Additionally, constructing the network and identities based on a static snapshot from 2018 introduces potential biases, as both network structures and user demographics evolve over time . The use of Twitter Decahose data, which includes only 10% of tweets, results in an incomplete and sparse network, particularly when relying on a mention-based network structure.

In reality, platform algorithms and temporal dynamics significantly influence hashtag diffusion. As such, the proposed model represents a simplification of complex social behaviors. Future work should aim to incorporate these factors to better reflect real-world diffusion processes.

The findings of Figure 1 (for growth) are particularly interesting, as both the CMI scores for the Network-only and Identity-only models are negative, while the Network+Identity model achieves a positive score. This outcome is counterintuitive and should be discussed to better understand the underlying reasons.

I find it slightly confusing that identity-related features are used both in constructing the CMI and as predictors in the regression analysis. This creates a circular logic where identity-related metrics influence the CMI score, and the regression then "finds" that these same metrics improve model performance.

In Section 7, The Optimal Combined Model, while theoretically interesting, has no practical utility as it relies on post-hoc knowledge that is unavailable in real-world applications. Additionally, the low accuracy (0.44) of the Predicted Combined Model suggests that the features used to predict the best model are insufficient to reliably differentiate between the underlying mechanisms (network, identity, or both).

**Questions:**

Any responses to the concerns mentioned above are welcome.

**Ethics Review Flag:**

Yes

**Reviewer Confidence:**

2: The reviewer is willing to defend the evaluation, but it is likely that the reviewer did not understand parts of the paper

**Scope:**

3: The work is somewhat relevant to the Web and to the track, and is of narrow interest to a sub-community

---

### Official Review · Reviewer_MMzn · 2024-12-02

**Novelty:** 6
**Technical Quality:** 6

**Review:**

The paper explores how hashtags diffuse on social media platforms by combining network and identity factors into a predictive model. The study uses an agent-based simulation to evaluate the adoption of 1,337 hashtags over 10 years of Twitter data. It introduces three models, Network-only, Identity-only, and Network+Identity, to analyze their performance in replicating real-world cascades. The results show that the combined Network+Identity model provides better predictions of cascade size, adopter composition, and growth patterns compared to simpler models. The authors also introduce a Composite Cascade Match Index (CMI) to evaluate model performance across diverse diffusion metrics and analyze contextual factors like topic and geographic alignment.


Strengths:

- The paper combines network and identity effects, offering a novel perspective on diffusion mechanisms beyond traditional single-factor models.
- A large dataset of hashtags spanning a decade ensures a comprehensive analysis, seems that capturing diverse patterns of hashtag diffusion.
- The CMI metric provides a helpful way to assess how well the models work across different aspects of diffusion.
- The paper analyze how hashtag topics (e.g., sports, social issues) influence model performance adds depth and practical value to the findings.
- The paper is well-organized and generally easy to follow from start to finish.


Weaknesses:

- The agent-based model lacks sufficient explanation of how agents decide to adopt a hashtag. For example, The process of updating adoption probabilities over time is unclear—do agents dynamically adjust their likelihood of adoption based on changing network exposure or is this static? The description of how the timestep-based diffusion evolves (e.g., how neighbors' influence propagates) needs more explicit details or pseudocode for better transparency.
- Equation (1) calculates p_{iht} (the probability of adoption) as proportional to a combination of factors, including network influence, identity alignment, and stickiness. However, The paper does not explain how the raw value is transformed into a valid probability within the range of [0, 1]. Without normalization, the values could exceed this range or fall below zero, making them invalid as probabilities. This omission leaves the implementation unclear and raises concerns about the mathematical soundness of the model.
- The model does not account for the role of content characteristics (e.g., semantic appeal, emotional resonance, or clarity of hashtags), which are often critical in determining whether a hashtag resonates with users. The influence of platform algorithms (e.g., trending algorithms or personalized recommendations) is ignored, even though these are known to significantly affect the visibility and adoption of hashtags. Omitting these factors may limit the generalizability and real-world applicability of the results.
- The classifier designed to select the best model (Network-only, Identity-only, or Network+Identity) for a given hashtag achieves only 44% accuracy, which is only marginally better than random guessing.
- The study is based entirely on Twitter data, and the findings may not extend to other platforms with different network structures, interaction mechanisms, and user demographics (e.g., Instagram, TikTok, or Facebook). A discussion of how these results might translate to different social media ecosystems is missing, leaving the applicability of the model unclear in broader contexts.
- The paper does not provide an in-depth analysis of where and why the Network+Identity model fails to outperform simpler models. For instance, Are there specific hashtag topics, cascade sizes, or adopter demographics where the combined model underperforms? Understanding these failure modes could help refine the model and guide future research.
- [Minor] Identity alignment (δih) is measured only by demographic proximity, such as geographic location or race, which oversimplifies the multi-faceted nature of identity. Real-world identity is often shaped by cultural, behavioral, and interest-based factors that are ignored here.

**Questions:**

How do the agents decide to adopt a hashtag?

How robust is the random forest classifier that seems to be close to just choosing a single model? What is the purpose of this line of experiments?

**Reviewer Confidence:**

3: The reviewer is confident but not certain that the evaluation is correct

**Scope:**

4: The work is relevant to the Web and to the track, and is of broad interest to the community

---

### Official Review · Reviewer_KXgt · 2024-12-02

**Novelty:** 5
**Technical Quality:** 6

**Review:**

Overall this paper uses an innovative method- agent based modeling- to investigate the ways in which hashtags diffuse on Twitter. Hashtag virality is a particularly prevalent feature of online discussions and how hashtags snowball into mass usage is worth investigating. The potential impact of this work lies in the information diffusion and the web-based communication research tracks. There are a number of strengths in this study which are as follows:

Pros:
- There are multiple research outputs in this study aside from research findings, including a network cascades dataset, alongside new cascade model evaluation framework and predictive model.
- The author(s) provide detailed explanations of all modeling choices and formula breakdowns
- The author(s) ensure internal validity via their rigorous hashtag pre-processing as state in Appendix D to isolate the effects of novel and organic viral hashtags

Cons
- I am concerned that the significance of the findings of this study are overfit. Despite the multi-factor approach, including the multiple demographic categories of agent identity (race, SES etc.) cultural diffusion online is influenced by a broader array of social, psychological, and contextual factors (e.g., platform algorithms, emotional content, influencer presence). Though it is outside the scope of this paper ignoring these could limit the model’s comprehensiveness and predictive accuracy.
- Based on the above concern the practical implications of these findings is also a bit precarious?
The remainder of the concerns are outlined in the questions section.

**Questions:**

- Does the change in Twitter character limits in 2017 impact hashtag use in any way and thus influence these results? For example, does the character limits encourage the use of hashtags since there is limited space to convey a message?
- How do you distinguish the cascading effects of identity from networks? It would assume demographic characteristics like race would preclude or determine your network to begin with. For example in your example of Black Twitter, being Black is likely what puts you in Black twitter networks to begin with not the other way around. In that way network and identity are synonymous. So would evaluating a model with just network effects be impossible because using salient hashtags from the Black twitter community is inherently also assuming Black identity...not for every case but for most? This is discussed a bit on line 744 but could the author(s) please clarify.
- Regarding "stickiness" or the constant of proportionality (𝑆ℎ) can the author(s) clarify a bit more what this represents? Could stickiness be more explicitly tied to specific behaviors or characteristics in social media contexts, such as how it reflects user engagement, virality, or content retention.
- Also does performing grid searches with only one run of the model per value may lead to variability or noise in the results?

**Reviewer Confidence:**

3: The reviewer is confident but not certain that the evaluation is correct

**Scope:**

3: The work is somewhat relevant to the Web and to the track, and is of narrow interest to a sub-community

---

### Official Review · Reviewer_DSp6 · 2024-12-03

**Novelty:** 3
**Technical Quality:** 3

**Review:**

Summary:
This paper explores the diffusion of hashtags on Twitter and examines how network and identity independently and interactively influence cascade properties. The authors simulate diffusion of 1337 hashtags across a network of 3 million nodes that was constructed from mutual mentions. They analyze cascade properties via a ten-factor framework, encompassing dimensions like popularity, growth, and adopter composition. Identity is inferred using demographic data from congressional district census statistics and geolocation. Their findings suggest that a combined network + identity model generally performs best, but specific properties, such as growth, popularity, or adopter composition, are better predicted by models that isolate network or identity effects. The paper proposes a dynamic model selection approach using random forests, which outperforms a baseline of always selecting the combined model.

General Comments:
Generally, I think this paper brings up a really interesting point of incorporating identity with well-established and heavily used network metrics. Identity, as mentioned in this paper, does play a role in media engagement and linguistic choice. This paper also uses a really large dataset that spans over a decade of collection (2012-2022) leveraging the decahose, which is a great longitudinal dataset for a study like this on hashtag neologism. I also appreciate that the authors breakdown cascades into different features, acknowledging that network/identity features can contribute disproportionately to different features of cascades. However, I do find that there are a number of issues with the paper that I would like to be addressed before I can confidently recommend this paper be accepted -- namely, I am the most concerned about some of the assumptions that are made when tagging users’ ‘identity’.

Generally, this study brings forward a compelling discussion on the interplay between identity and network features. Identity is difficult to measure on social media platforms including Twitter, despite identity being an important role in media engagement and linguistic diffusion. The use of a decade-long dataset (2012-2022) via the Decahose also provides a strong foundation for analyzing longitudinal trends in hashtag neologism. I also commend the authors for recognizing the heterogeneous contributions of identity and network across cascade dimensions. However, I have several critical concerns about the methodology, specifically regarding the assumptions underlying identity tagging, the generalizability of these measures and potential campaign activity.
Assumptions and Generalizations in Identity Tagging
The use of congressional district-level census statistics to infer individual identity raises significant concerns. Users within a district may exhibit substantial heterogeneity in race, socioeconomic status and/or political affiliation. Generalizing district-level data fails to account for these intra-district variations, particularly in diverse and contested regions. This limitation may undermine the validity of the identity metrics for certain segments of the population.
Twitter’s demographic skews (younger, more left-leaning users per Pew Research, 2019) further complicate these generalizations. Mapping a district’s demographic composition onto individual users within such a skewed platform demographic risks introducing systemic biases.
Temporal factors are not thoroughly addressed. Given that this dataset encompasses a decade's worth of data, user locations and the demographics of districts may have shifted, and user political alignments may also have changed over time. This has become more glaringly evident over the last 5-10 years, with left-leaning/right-leaning identities changing and evolving.

Coordinated campaigns versus Organic growth
Automated accounts or coordinated campaigns could significantly affect the diffusion of hashtags, particularly in the early stages. Were these kinds of users (especially bots) taken into account in this work?
Was there a differentiation made between campaigns that pushed neologism (political campaigns, sports etc.) versus non-campaigns and more organic neologism diffusion? I understand that the hashtags were categorized based on topic, but there may be a difference between coordinated versus organic activity even within a topic.

The study’s contributions to understanding hashtag cascades through network and identity perspectives are valuable. However, I have fairly major concerns surrounding the identity metrics that need to be addressed, especially since these are district-level metrics being applied to individual-level metrics which can introduce significant biases. Clarifying these points and addressing potential limitations will significantly strengthen the robustness, trustworthiness and impact of the findings.

I understand that addressing these fundamental issues—such as the representativeness and accuracy of identity metrics—may require substantial revisions that could exceed the time available in WWW’s review cycle.

*Strengths*
- Important integration of network and identity metrics:
The paper combines network and identity features which offers an important perspective on modeling hashtag diffusion.
- Dataset is massive and spans a decade:
The authors have access to a decade worth of Decahose data, which is a fantastic foundation for analyzing longitudinal data in hashtag neologism, adoption and diffusion.
- Granular breakdown of cascade features:
The authors acknowledge the different properties of a cascade, and suggest a 10 factor framework (e.g. growth, popularity and adopter composition) to characterize cascades.



*Weaknesses*
- Generalizability of Identity Metrics:
Using congressional district-level census data to infer individual identity is concerning because it overlooks itra-district heterogeneity and introduces potential biases -- particularly for diverse or contested (swing) districts.
- Temporal shifts in identity metrics not addressed:
The paper does not adequately address how evolving demographics, user locations and/or political alignments over the dataset’s decade-long span might impact the accuracy or representativeness of using current identity metrics in cascade modeling. Some of these assumptions and generalizations seem a bit generous.
- Coordinated vs. organic activity:
The authors should include discussion of how coordinated campaigns versus organic neologism diffusion may play a role in predictive power of identity versus network features. The authors divide hashtag cascade simulations based on topic, but there can and are these different kinds of activity within the same topic.

**Questions:**

see above

**Reviewer Confidence:**

4: The reviewer is certain that the evaluation is correct and very familiar with the relevant literature

**Scope:**

3: The work is somewhat relevant to the Web and to the track, and is of narrow interest to a sub-community